# '12-Angry LLMs' - Divergence from Deliberation as Signal for Complex Stance Detection

## Abstract

This study proposes a multi-LLM jury model for complex stance detection that uses model disagreement and rationale traces as signals of uncertainty and interpretability. A diverse panel of LLM jurors first produces independent stance judgments and rationales; uncertain cases are then revisited via anonymized peer summaries for self-refinement. The resulting jury rationales are used as auxiliary input to downstream stance classifiers, including a fine-tuned Judge trained against human expert gold labels. On RUStance-2023, the fine-tuned Judge achieves the strongest performance among evaluated configurations. Applied end-to-end to PStance, SemEval-2016, and GWStance using each dataset's native label schema, the model shows dataset-dependent gains, with the clearest advantage on RUStance-2023. Overall, the results suggest that multi-perspective rationale supervision can improve stance detection in difficult settings while also exposing disagreement signals that are useful for auditing uncertainty. The dataset and code will be released to support future research on divergence in stance detection.

## 1 Introduction

The task of identifying whether a text is in favor of or against a specific target issue or claim is known as stance detection. It offers a systematic way to analyze viewpoints (Tao et al., 2024). This task has broad applications; for example, in the Russia-Ukraine conflict, social media has been overwhelmed with competing narratives, and stance classification helps map how users align with or oppose key claims, thereby assisting in the detection of propaganda and misinformation (Lavrouk et al., 2024; Gorrell et al., 2019; Conforti et al., 2018). Prior work has demonstrated the viability of stance classification as an effective tool for identifying misinformation across languages and communities (Lavrouk et al., 2024). More generally, automatic stance detection plays an important role in information retrieval, content moderation, and text summarization, where understanding an author's standpoint provides valuable contextual insight (Mohammad et al., 2016; Conforti et al., 2020; Carnot et al., 2023).

Determining a stance on contentious issues is often a subjective task, and annotator divergence is a well-recognized challenge. Even expert human annotators frequently disagree on stance labels owing to ambiguous language, differing cultural backgrounds, or personal biases (Li et al., 2025). Traditionally, such disagreements are treated as noise to be resolved via majority voting or averaging. However, emerging research argues that disagreement is not merely noise but a signal of genuine differences in perspective (Fleisig et al., 2023; Uma et al., 2021). For example, studies have found that annotator disagreements in labeling hate speech or rumor veracity often stem from systematic factors, such as annotators' demographics or political leanings, rather than random error (Fleisig et al., 2023). In stance datasets, relying solely on majority votes can yield inconsistent or contradictory gold labels for similar content, underscoring the limitation of blurring out minority viewpoints (Li et al., 2025). This divergence is not restricted to humans; Large Language Models (LLMs) used as annotators or classifiers may also produce varying judgments depending on model design or prompting. A single AI "judge" can exhibit inherent biases (Zheng et al., 2023; Liu et al., 2023); hence, stance analysis using LLMs demands methods that account for multiple interpretations rather than one-size-fits-all predictions (Jain, 2025).

In this study, we address annotator disagreement and propose a new approach to leverage it for better stance detection. Drawing inspiration from deliberative juries (as dramatized in the movie '12 Angry Men'), we introduce a jury of LLMs in stance detection task. In our model, multiple independent LLMs act as 'jurors' who each cast a stance judgment before their votes are deliberated upon and aggregated to produce a final label by a fine-tuned expert LLM that acts as a 'judge'. In uncertain cases, each juror is re-prompted once after receiving an anonymized summary of the vote distribution and representative rationales. A point to note is that this step is procedural peer-context self-refinement, not open-ended debate or human-like persuasion. The resulting traces are then used by a fine-tuned Judge model trained against human expert gold labels. This multi-LLM strategy is designed to generalize the annotation process beyond any single annotator's biases, leveraging diversity to improve reliability. We evaluated this methodology on our RUStance-2023 (Anonymous, 2026) data set alongside publicly available data sets. The deliberation and jury-based labeling are added to extend the RUStance-2023 data set. In summary, our contributions are as follows:

1. **Jury of LLMs.** We adapt a jury-of-LLMs model for stance detection, using a panel of twelve LLMs to produce independent labels and rationales. Inspired by the deliberative process in the film *12 Angry Men* and the formal logic of courtroom juries, we treat each LLM as an independent evaluator and aggregate its votes through structured multi-round deliberation. This approach provides a methodology for labeling (or classifying) subjective data, reducing the influence of individual model bias, and capturing disagreements as signals of underlying uncertainty. Note that twelve is a fixed experimental design choice and a mnemonic for the multi-juror setting, not a theoretically justified optimum.

2. **Benchmarking Across LLMs, and Classical Models.** We conducted extensive experiments comparing the stance classification performance when trained on expert gold labels. We further evaluated multiple modeling approaches, including multilayer perceptrons (MLPs), zero-shot LLMs, few-shot LLMs, and fine-tuned LLMs. This study provides the first systematic comparison of LLM and jury-based annotations for stance detection.

3. **Russia–Ukraine Stance and Deliberation Dataset.** We will release a dataset of social media posts on the Russia–Ukraine conflict, annotated by a jury of 12 LLMs (Juror-RoundA Dataset). In addition to the stance labels, the dataset includes detailed records of the jury's deliberation process (Juror-RoundB Dataset), enabling researchers to study the final labels and the dynamics of LLM disagreement and consensus.

Overall, our work introduces a general deliberative annotation and evaluation model for stance detection. By combining human expert gold labels with LLM-based jury deliberation, we provide a broader benchmark for studying disagreement as a cue in stance detection.

This paper begins with a discussion of related work in Section 2. Section 3 presents the methodology in detail. Section 4 provides the experimental setup, including the baseline models, LLM configurations, and aggregation strategies. The experimental findings are reported and analyzed in Section 5, highlighting the quantitative performance and qualitative insights into disagreements. Finally, Section 7 concludes the paper with reflections on contributions and outlines possible directions for future work.

## 2 Related Work

Stance detection has been an active area of research in NLP for the past decade. Early benchmarks such as 'SemEval-2016 Task 6: Detecting Stance in Tweets' introduced datasets of tweets labeled as Favor, Against, or Neither towards specific targets (Mohammad et al., 2016). Subsequent datasets expanded the scope, including P-Stance with over 21,000 political tweets labeled for stance towards U.S. politicians (Li et al., 2021) and COVID-19-Stance, capturing pandemic-related debates on social media (Glandt et al., 2021). Methods have evolved from lexical and sentiment-based classifiers to deep neural models and, more recently, to fine-tuned transformers such as BERT and RoBERTa, which have become state-of-the-art on benchmarks such as SemEval and P-Stance (Li et al., 2021). Despite these advances, generalization across unseen targets remains challenging, and stance detection relies heavily on the quality and reliability of annotations.

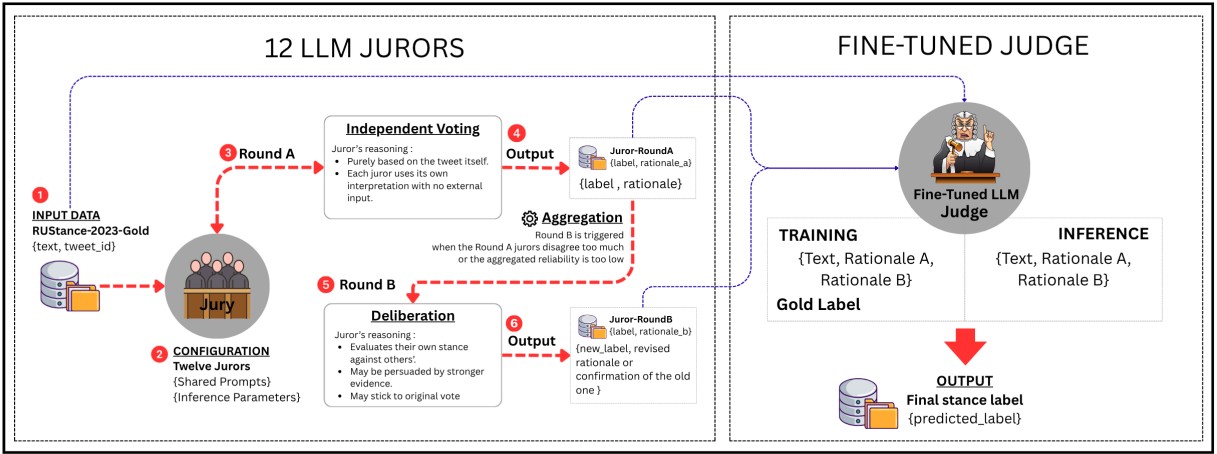

Figure 1: Process Flow of the Deliberative Jury of LLMs for Stance Detection.

A recurring difficulty in stance research is the subjectivity of the annotation. Unlike factual classification tasks, stance often depends on ambiguous cues and the annotator's interpretation, which leads to significant annotator disagreement. For example, in the SemEval-2016 stance dataset, tweets were double-annotated by a pool of eight annotators. The organizers reported an initial inter-annotator agreement of 73% (percent agreement) across all instances (Mohammad et al., 2016), reflecting the difficulty and ambiguity in many cases (e.g., sarcasm or context-dependent opinions). To ensure a reliable evaluation set, the task designers filtered out tweets with low consensus: only instances where at least five of eight annotators agreed ($\geq 60\%$ agreement) were included in the official Stance Dataset. This increased the agreement on the released data to 81.9% (Mohammad et al., 2016), but it also meant that highly controversial or ambiguous examples were set aside as too difficult. Similarly, other stance annotation efforts have reported moderate levels of agreement. For example, recent studies note that Cohen's $\kappa$ between annotators is often around 0.4–0.5 for stance-like subjective tasks, indicating only "moderate" agreement by conventional scales (Schiller et al., 2021). These results underscore that identifying stance can be inherently subjective; different people may legitimately interpret the exact text in favor of or against a target based on their background knowledge or beliefs. Low annotator agreement raises concerns regarding the reliability of stance labels and the validity of treating any single "ground truth" as authoritative. Conventionally, NLP datasets resolve conflicting annotations by majority vote or averaging, thereby collapsing disagreements into single labels. This practice is widespread in subjective annotation tasks (Mostafazadeh Davani et al., 2022). However, as Mostafazadeh Davani et al. (2022) argue, disagreements often "reflect annotators' individual perspectives and value" and may "encode fine distinctions that are typically overlooked" when a consensus label is imposed.

Recently, LLMs have opened new possibilities for automating annotations and LLMs are used as annotators or crowdworkers instead of humans. Gilardi et al. (2023) demonstrated that GPT-based models can outperform crowdworkers in accuracy and inter-coder reliability on multiple text annotation tasks, including stance classification. However, LLM outputs are sensitive to prompt wording and decoding randomness, raising concerns regarding consistency. To address this, self-consistent decoding (Wang et al., 2023) aggregates multiple reasoning paths from the same model to produce more reliable outputs, echoing ensemble strategies that have long been used in human annotation. Taking this ensemble approach a step further, researchers have begun experimenting with multi-agent LLM models in which several models debate, critique, or vote before making decisions. Zhao et al. (2024) shows that multi-LLM debate can significantly improve factual and logical accuracy, while Choi et al. (2025) reports that discussion phases before voting lead to stronger consensus. These approaches parallel human committees, suggesting that ensembles of models may yield more reliable outcomes than those of single annotators. In contrast to prior work, which typically attempts to minimize or eliminate disagreements, our study treats divergence as a meaningful signal. By introducing a jury of LLMs, we preserved both agreement and disagreement. Unlike the multi-agent debate model

that aims to produce a single, superior consensus, our method is designed to output the full spectrum of divergence, including structured minority reports and hung jury statistics. This enables a systematic study of when models converge and split and what these patterns reveal about subjectivity and ambiguity in stance detection.

## 3 Methodology

The overall methodology is illustrated in Figure 1. The methodology is divided into two parts. First jury deliberation dataset creation and second stance detection using fine-tuning the LLM model as the judge.

### 3.1 Jury of LLMs

We design a jury-of-LLMs process in which twelve LLMs act as independent jurors. Each juror interprets the same input text, which is a short tweet, and then produces a stance label, confidence score, and rationale for their choice. The choice of twelve balances diversity and efficiency and echoes the deliberative design popularized by *12 Angry Men* and the formal logic of courtroom juries. The jury comprised twelve large language models sourced from different providers and architectures to maximize the diversity of perspectives. The different LLMs used are listed in Table 1. The table shows that the composition ensures coverage across instruction-tuned general-purpose models (e.g., GPT-4o-mini, Claude-3.5, Gemini-2.5), specialized or hybrid models (e.g., Qwen3-Coder, WizardLM, Jamba-Mini), and large open-weight architectures (e.g., Llama-3.3-70B, Nemotron-49B, Mistral). By mixing providers and scales, the jury is deliberately designed to capture a wide range of training data biases, reasoning styles and annotation behaviors.

Table 1: Roster of LLM jurors used in the multi-round deliberation.

| ID | LLM Model Variant | Parameter |
|----|-------------------|-----------|
| J1 | gpt-4o-mini-2024-07-18 | undisclosed |
| J2 | gemini-2.5-flash | undisclosed |
| J3 | claude-3.5-sonnet | undisclosed |
| J4 | grok-4 | undisclosed |
| J5 | llama-3.3-70b-instruct | 70 B |
| J6 | llama-3.3-nemotron-super-49b-v1 | 49 B |
| J7 | deepseek-chat-v3.1 | 671 B, 37 B active |
| J8 | qwen3-coder | 480 B, 35 B active |
| J9 | glm-4.5 | 355 B, 32 B active |
| J10 | jamba-mini-1.7 | 52 B, 12 B active |
| J11 | mistral-nemo | 24 B |
| J12 | wizardlm-2-8x22b | 22 B |

### 3.2 Multi-Round Deliberation and Aggregation

The jury of LLMs is organized into two structured annotation rounds designed to capture independent judgments and optional peer-context self-refinement under uncertainty. To encourage transparent reasoning and enable analysis beyond the final label, each juror is instructed to produce a brief rationale first. The rationale captures the model's explanation for its stance decision (e.g., citing explicit support for Ukraine, detecting sarcasm, or noting ambiguous language). These rationales are later used as evidence in the deliberation phase and as sources of minority reports.

Each juror is then zero-shot prompted with a rationale-then-label instruction: the model first explains its reasoning and only afterward emits a single stance label from the predefined stance label set. Prompts are kept identical across jurors and rounds (except for Round B, where contextual vote distributions and selected rationales are added) to avoid confounding.

1. **Round A (independent votes).** In the first round, each juror independently assigns a stance label without exposure to the views of other models. Each model receives the same input text and a standardized system prompt that describes the stance labeling schema, which includes six possible categories: Pro-Ukraine, Pro-Russia, Pro-NATO, Anti-NATO, Neutral, and Unclear. Each juror outputs a JSON object containing: (i) **label**: predicted stance category, (ii) **confidence**: self-assessed confidence probability (0–1), (iii) **uncertain**: flag for ambiguity or sarcasm, (iv) **rationale**: brief textual justification, and (v) **evidence**: key phrases or sub-strings supporting the decision.

   This produces twelve parallel judgments that reflect the unmediated diversity of perspectives across models. Thus, Round A serves as a baseline measure of individual variance, analogous to the initial polling of a human jury.

2. **Round B (deliberation with selected rationales).** After Round A, each juror independently assigns a stance label and provides a rationale. Round B is activated only when the initial jury decision is uncertain. In our implementation, uncertainty is estimated from the Round A vote distribution using three signals: vote entropy, vote margin, and reliability. Entropy measures how dispersed the juror votes are across labels; high entropy indicates disagreement. Vote margin measures the difference between the most frequent and second-most frequent labels; a small margin indicates that the winning label is weak. Reliability measures the proportion of jurors supporting the final aggregated label. When entropy is high, the margin is low, or reliability is below a predefined threshold, the system triggers Round B deliberation. In Round B, jurors are exposed to the Round A vote distribution and representative rationales, including both majority and minority reasoning. They are then asked to reconsider their stance. This second round is intended to test whether disagreement can be reduced after jurors observe alternative justifications. If the jury is already confident in Round A, Round B is skipped to reduce latency and cost.

   Reliability is measured as the level of agreement between the jury and the final aggregated decision. Under majority voting, reliability is computed as the fraction of jurors whose vote matches the final majority label. For example, if four out of six jurors vote for the final label, reliability is $4/6 = 0.667$. A reliability score close to 1.0 indicates strong agreement, while a lower score indicates uncertainty or disagreement. Reliability is therefore not the same as correctness against the gold label; it is an internal confidence measure based on jury agreement. For aggregation, we report both majority voting (MV) and Dawid-Skene (DS) aggregation. Majority voting assigns the final label to the class that receives the most juror votes. Dawid & Skene (1979) is a probabilistic aggregation method that estimates both the latent true label and the reliability of individual annotators. In our setting, each LLM juror is treated as an annotator. The method iteratively estimates how likely each juror is to produce each label given the latent true label, then uses these estimated juror-specific confusion patterns to infer a final label. This allows the aggregation process to account for the fact that some jurors may be more reliable than others or may systematically confuse certain stance categories.

### 3.3 The Judge: Deliberation, Gold Label and Inferencing

The Judge is trained as a supervised stance classifier using expert-labeled examples. For each tweet, we concatenate the original tweet text with the jury-generated rationales from Round A and Round B, producing a text-plus-rationale input. The target output is the expert stance label from the gold dataset. An expert stance label is the final consensus annotation provided by human domain experts ( Anonymous (2026)). Unlike jury aggregation methods, which combine juror votes through majority voting or Dawid–Skene, the Judge uses the semantic content of the jurors' rationales as evidence. During inference, the trained Judge receives an unseen tweet and its associated deliberation traces, then predicts one of the six stance labels. Thus, the Judge is not a vote aggregator, but a supervised classifier that tests whether deliberation rationales provide useful signal beyond the jury vote distribution.

# 4 Experiments

We conducted a systematic evaluation of stance classification under three experimental configurations. The first configuration is the MLP as a judge, which includes classical neural baselines that use text embeddings and text embeddings combined with jury-provided rationales. The second Configuration, LLM (few-shot) as a judge, and finally, the Fine-Tuned LLM as a judge. All these experiments are tested on human gold standard data with the proposed multi-LLM juror deliberation dataset.

## 4.1 Dataset Alignment

We used the RUStance-2023-Gold dataset (300 posts) as an expert-labeled dataset (Anonymous (2026)). To train the meta-judging model, we aligned the multiround jury deliberation outputs with the expert-labeled ground truth to form a structured supervision dataset. The alignment process integrates the rationales from Rounds A and B for all jurors, along with the corresponding expert stance label for each post. Specifically, for each tweet_id, the script extracts every juror's explanatory rationale from both rounds, cleans the text, and concatenates them to form a composite reasoning input. This composite string represents the collective deliberation context of the post. The resulting dataset pairs (Text + Combined Rationales) are used as inputs, with the expert label as the target.

This alignment ensures that the fine-tuned Judge model learns to map the jury's reasoning traces to gold-standard human annotations, rather than simply reproducing the jury majority vote. By exposing the model to both initial Round A rationales and reconsidered Round B rationales, the training data captures a richer deliberative context, including alternative justifications, revised reasoning, and cases of disagreement or ambiguity. As a result, the Judge can learn patterns in the combined text-and-rationale input that predict expert decisions, thereby serving as a supervised bridge between collective model reasoning and authoritative human annotation.

In addition to RUStance-2023, we applied the same end-to-end experimental pipeline to three additional stance detection datasets: PStance (Li et al., 2021), SemEval-2016 Task 6 (Mohammad et al., 2016), and GWStance (Luo et al., 2020). For each dataset, the jury generated Round A and Round B rationales on that dataset's own texts using the dataset's native label schema: favor/against for PStance, FAVOR/AGAINST/NONE for SemEval, and agree/disagree/neutral for GWStance. For each dataset, the combined Round A/B rationales were concatenated with the original text and used as input to the corresponding Judge model and baseline configurations. All downstream configurations, including the fine-tuned Judge, were trained and evaluated on the same dataset under 4-fold stratified cross-validation. Few-shot exemplars were drawn only from the training split of each fold.

## 4.2 Experiment Configuration

To evaluate the effectiveness of our proposed jury-of-LLMs model, we compared it with a range of models, including classical neural architectures, zero- or few-shot prompting with large language models, fine-tuning on each dataset's training folds with gold stance labels, and multi-model jury aggregation. These baselines provide a comprehensive view of the performance trade-offs between accuracy, stability, and interpretability.

Table 2: Comparison of experimental setup for different LLM configurations.

| Aspect | Zero-Shot LLMs | Few-shot LLMs | Fine-tuned LLMs |
|---|---|---|---|
| **Model** | gpt-4.1-mini (OpenRouter) | gpt-4.1-mini (OpenRouter) | gpt-4.1-mini (April 2025 fine-tuned) |
| **Input format** | Text (with/without rationales) | Text + 3 curated examples per class (with/without rationales) | Text + aggregated jury rationales |
| **Training protocol** | No task-specific training | No task-specific training | Fine-tuning on RUStance-2023 |
| **Prompting strategy** | Direct classification into 6 stance labels | Exemplars prepended as in-context demonstrations | Finetuning with gold stance labels |
| **Repetition / Voting** | 3 repeated calls; majority vote | 3 repeated calls; majority vote | Deterministic outputs (single label per instance) |
| **Evaluation split** | 4-fold cross-validation | 4-fold cross-validation | 4-fold cross-validation |

### 4.2.1 Configuration 1: MLP as Judge

We first established a classical baseline using a multilayer perceptron (MLP) trained on the tweet text and jury rationales. Each tweet was encoded into a 768-dimensional vector using the `all-mpnet-base-v2` sentence transformer, while the juror rationales and deliberation texts were separately embedded and averaged into an additional 768-dimensional vector. These two representations were concatenated to form a 1,536-dimensional input feature vector. The MLP consisted of two hidden layers (512 and 128 units) with ReLU activations and dropout regularization. Training was performed using Adam optimization for 20 epochs, using a cross-entropy loss function with class weights to address label imbalance. The evaluation was conducted under a 4-fold stratified cross-validation, reporting accuracy, macro-F1, and Cohen's $\kappa$. Predictions below a confidence threshold of 0.5 were reassigned to the *Unclear* class to handle uncertainty. This setup enabled us to test whether jury rationales contribute to a machine-usable signal when integrated with textual embeddings.

### 4.2.2 Configuration 2: LLM (few shots) as Judge

Building on this, we evaluated large language models (LLMs) as stance classifiers without task-specific training. All experiments used gpt-4.1-mini, accessed via the OpenRouter API with deterministic decoding. In the few-shot setting, we prepended two to three curated exemplars per class from the RUStance-2023 dataset, providing additional context. Each classification was repeated three times, and the majority label was taken as the final prediction. Rationale for Rounds A and B included assessing whether exposing the models to prior explanations improved consistency.

### 4.2.3 Configuration 3: Fine-Tune LLM as Judge

Fine-tuned gpt-4.1-mini (April 2025 ) on RUStance-2023 using concatenated tweet text and aggregated jury rationales as input. Fine-tuning was evaluated under complementary protocols. We performed a four-fold stratified cross-validation to test the robustness. The model was constrained to produce one of the six stance labels. Performance is measured using accuracy, the weighted F1 score, and confusion matrices. This experiment examined whether fine-tuning with jury rationales yields more stable and reliable predictions than prompting alone does. The configurations of the LLMs are listed in Table 2.

## 5 Results and Analysis

This section presents the empirical results from our experimental model, comparing classical machine learning baselines, zero and few-shot LLMs, a jury of LLMs aggregation, and fine-tuned LLMs. Our analysis focuses on three dimensions: predictive performance, reliability across classes, and output interpretability.

### 5.1 Overall Model Performance Results

Table 3: Macro-F1 Performance Across Datasets

| Model | RUStance | PStance | SemEval | GWStance | Avg Macro-F1 |
|---|---|---|---|---|---|
| MLP (Text) | 0.428 | 0.675 | 0.628 | 0.885 | 0.654 |
| MLP (Text + Rationale AB) | 0.475 | 0.819 | 0.697 | **0.929** | 0.730 |
| Zero-shot LLM (Text) | 0.454 | 0.686 | 0.733 | 0.790 | 0.666 |
| Zero-shot LLM (Text + Rationale AB) | 0.611 | 0.922 | 0.705 | 0.790 | 0.757 |
| Few-shot LLM (Text + Rationale AB) | 0.647 | 0.943 | **0.736** | 0.783 | 0.777 |
| Jury MV (Round A) | 0.603 | 0.363 | 0.693 | 0.800 | 0.615 |
| Jury DS (Round A) | 0.621 | 0.423 | 0.676 | 0.796 | 0.629 |
| Jury MV (Round B) | 0.617 | 0.429 | 0.690 | 0.777 | 0.628 |
| Jury DS (Round B) | 0.624 | 0.438 | 0.657 | 0.778 | 0.624 |
| Finetuned LLM Judge (Text + Rationale AB) | **0.815** | **0.946** | 0.693 | 0.911 | **0.841** |

Table 3 presents the overall Macro-F1 performance of all model configurations across four stance datasets: RUStance-2023, PStance, SemEval, and GWStance. Several important trends emerged from this compar-

ison. First, incorporating jury deliberation rationales generally improves performance across several configurations, particularly for RUStance-2023 and PStance. For example, the MLP improves from 0.428 to 0.475 on RUStance and from 0.675 to 0.819 on PStance when rationales are added. Similarly, zero-shot prompting improves from 0.454 to 0.611 on RUStance and from 0.686 to 0.922 on PStance when augmented with rationale. These suggest that jury-generated rationales are especially useful for datasets where stance detection requires implicit reasoning and contextual interpretation. However, the benefits of rationales are dataset-dependent. On SemEval, zero-shot rationale augmentation reduces Macro-F1 from 0.733 to 0.705, and on GWStance, it does not improve over the zero-shot text-only setting. Conversely, MLP with rationale augmentation achieves the strongest GWStance result, with a Macro-F1 of 0.929 and finetuned Judge produced second best result with 0.911. These mixed results indicate that deliberative rationales are not uniformly beneficial across all stance datasets, and their usefulness depends on the label schema, dataset characteristics, and the degree of implicit stance expression.

Across all datasets, the fine-tuned LLM Judge, trained on text and aggregated jury rationales, achieves the highest average Macro-F1: 0.841. It achieves the best results on two of four datasets: RUStance-2023 (Macro-F1: 0.815) and PStance (Macro-F1: 0.946). The Few-shot LLM performs best on SemEval, with a Macro-F1 of 0.736, while MLP with rationale augmentation performs best on GWStance, with a Macro-F1 of 0.929. Jury MV (Round A) obtains an average Macro-F1 of 0.615, while Jury DS (Round A) obtains 0.629. After Round B, Jury MV reaches 0.628, and Jury DS reaches 0.624. Jury MV/DS (Round A) denotes direct aggregation of the independent first-round juror labels using either majority vote (MV) or Dawid-Skene (DS). Jury MV/DS (Round B) denotes aggregation after the Round-B revisiting step: uncertain Round-A cases are shown an anonymized panel summary, jurors revise or reaffirm their labels, and the revised labels are then aggregated using MV or DS. Thus, Round A reflects independent voting, whereas Round B reflects voting after peer-summary deliberation. These results show that direct aggregation of juror votes is a meaningful baseline, but it is weaker in the conducted experiments. Thus, the Judge's performance is not simply the result of majority voting or Dawid-Skene aggregation over juror labels. Instead, the strongest gains come from using the jury-generated rationales as supervised input to the downstream Judge. Overall, the proposed Judge is the strongest configuration on average, but it is not uniformly the best across all datasets. The results suggest that jury-generated rationales can provide useful task-specific reasoning signals, particularly for complex or implicit stance-detection settings. At the same time, the variation across SemEval and GWStance shows that the effect of deliberation is also dataset-dependent.

## 5.2 RUStance-2023 Results

The detailed in-domain performance on the RUStance-2023 dataset (Table **??**) across various model configurations reveals a clear and consistent improvement with additional deliberative information and task adaptation. The classical MLP baseline achieved the lowest performance, with a Macro-F1 of 0.428 using text only, which increased to 0.475 when jury rationales were included, indicating that deliberation text contains useful semantic information even for traditional machine learning models. For LLM-based approaches, zero-shot prompting with text only achieved a Macro-F1 of 0.454. When Round A rationales are added, the performance increases substantially to 0.543, and further improves to a Macro-F1 of 0.611 when both Round A and Round B rationales are included. This demonstrates that the deliberation process provides additional reasoning signals that help the model better interpret stances in ambiguous or context-dependent posts. Few-shot prompting with deliberation rationales further improves performance to a Macro-F1 of 0.647, showing that combining rationales with in-context examples provides complementary benefits.

The best performance is achieved by the fine-tuned LLM Judge, which reaches a Macro-F1 score of 0.791. This suggests that aggregated jury rationales are particularly effective as supervision signals for fine-tuning, enabling the Judge model to learn how expert annotations relate to collective jury reasoning and effectively bridge the gap between raw model outputs and authoritative human labels. Overall, the stepwise improvement across these configurations indicates that jury deliberation rationales provide progressively more value as the model learns to interpret and utilize them.

Figure 2 shows that performance improvements are not only reflected in overall Macro-F1, but also in clearer class-level separation. The MLP baselines show confusion, especially between the *Neutral* and *Unclear* classes, whereas the few-shot LLM settings produce a stronger diagonal structure, indicating better

Table 4: Performance Comparison of Models on Stance Detection

| Model | Accuracy | Weighted F1 | Macro-F1 |
|---|---|---|---|
| MLP (Text) | 0.480 | 0.476 | 0.428 |
| MLP (Text + Rationale AB) | 0.513 | 0.510 | 0.475 |
| Zero-shot LLM (Text) | 0.471 | 0.454 | 0.454 |
| Zero-shot LLM (Text + Rationale A) | 0.546 | 0.543 | 0.543 |
| Zero-shot LLM (Text + Rationale AB) | 0.606 | 0.611 | 0.611 |
| Few-shot LLM (Text + Rationale AB) | 0.653 | 0.647 | 0.647 |
| Jury MV (Round A) | 0.603 | 0.603 | 0.603 |
| Jury DS (Round A) | 0.630 | 0.621 | 0.621 |
| Jury MV (Round B) | 0.617 | 0.617 | 0.617 |
| Jury DS (Round B) | 0.627 | 0.624 | 0.624 |
| Finetuned LLM Judge (Text + Rationale AB) | **0.81** | **0.80** | **0.791** |

alignment with the stance schema. Adding the Round A and B rationales further reduced off-diagonal errors, particularly for the more clearly expressed stance categories. The fine-tuned Judge model achieved the strongest diagonal dominance overall, with the remaining errors concentrated mainly in the inherently ambiguous *Neutral* and *Unclear* classes. This pattern supports the view that jury rationales are especially valuable for resolving difficult cases while ambiguity remains concentrated in weakly signaled posts.

## 5.3 Disagreement vs. Rationale Quantity Controls

These controls are designed to test whether performance improves simply because the model receives more chain-of-thought-style rationales, or whether structured disagreement among jurors provides an additional useful signal.

First, we evaluate a single-model twelve-rationale control on the RUStance-2023 dataset. In this setting, each example is paired with twelve sampled rationales generated by the same model, using the same label set and prompt format as in the jury condition. These rationales are then passed to the same downstream Judge. This control keeps the number of rationales fixed while removing cross-model disagreement. As shown in Table 5, the fine-tuned Judge achieves 0.597 accuracy, 0.566 Macro-F1, and 0.567 Weighted-F1. This result suggests that simply increasing the number of rationales from a single model does not reproduce the stronger gains observed in the disagreement-based jury setting.

Table 5: Single-model twelve-rationale control on RUStance.

| Method / Setting | Accuracy | Macro-F1 | Weighted-F1 |
|---|---|---|---|
| MLP (Text) | 0.530 | 0.486 | 0.527 |
| MLP (Text + Rationale AB) | 0.520 | 0.458 | 0.514 |
| Zero-shot LLM (Text) | 0.500 | 0.480 | 0.480 |
| Zero-shot LLM (Text + Rationale AB) | 0.523 | 0.510 | 0.510 |
| Few-shot LLM (Text + Rationale AB) | 0.593 | 0.593 | 0.593 |
| **Finetuned LLM Judge (Text + Rationale AB)** | **0.597** | **0.566** | **0.567** |

Second, we evaluate a length-matched consensus-rationale control on on the RUStance-2023 dataset. For each example, we generate one consensus rationale whose length is approximately matched to the concatenated jury rationales for that example. This gives the Judge a comparable amount of reasoning text while removing the explicit multi-juror disagreement structure. As shown in Table 6, the fine-tuned Judge achieves 0.740 accuracy, 0.725 Macro-F1, and 0.725 Weighted-F1. This indicates that longer, higher-quality consensus rationales are also beneficial. However, because this condition removes the disagreement structure, it allows us to distinguish the benefit of rationale quantity from the benefit of disagreement.

Table 6: Length-matched consensus-rationale control on RUStance. This setting keeps rationale quantity approximately fixed but removes the multi-juror disagreement structure.

| Method / Setting | Accuracy | Macro-F1 | Weighted-F1 |
|---|---|---|---|
| MLP (Text) | 0.507 | 0.494 | 0.507 |
| MLP (Text + Rationale AB) | 0.487 | 0.454 | 0.477 |
| Zero-shot LLM (Text) | 0.490 | 0.475 | 0.475 |
| Zero-shot LLM (Text + Rationale AB) | 0.487 | 0.489 | 0.489 |
| Few-shot LLM (Text + Rationale AB) | 0.570 | 0.569 | 0.569 |
| **Finetuned LLM Judge (Text + Rationale AB)** | **0.740** | **0.725** | **0.725** |

Overall, these controls show that rationale quantity alone does not fully explain the benefit of the jury-based setting. The single-model twelve-rationale control performs only modestly, despite receiving the same number of rationales. The length-matched consensus-rationale control performs substantially better, showing that extended, coherent reasoning is useful; however, it does not contain explicit cross-model disagreement. We therefore revise our interpretation: disagreement is not the only useful signal, but structured disagreement provides information beyond simply increasing the amount of rationale text.

## 5.4 Statistical Significance Testing

To evaluate whether the performance differences between the model configurations were statistically reliable, we conducted paired bootstrap resampling on the Macro-F1 scores. The results are presented in Table 7. The analysis shows that few-shot prompting with jury rationales significantly outperforms zero-shot prompting with rationales ($\Delta F1 = 0.0367$, $p = 0.0026$), indicating that in-context examples and deliberative rationales provide complementary benefits to the model. Furthermore, the fine-tuned Judge model significantly outperforms the few-shot configuration ($\Delta F1 = 0.0636$, $p = 0.0020$), demonstrating that aggregated jury rationales serve as effective supervision signals for training a specialized stance classifier.

We observe that incorporating jury deliberation rationales significantly improves zero-shot prompting performance compared with text-only zero-shot prompting ($\Delta F1 = 0.1557$, $p < 0.001$). This result indicates that deliberative rationales provide useful reasoning signals even without task-specific adaptation, enabling the model to better interpret ambiguous or context-dependent stance expressions. The improvement is consistent with the gains observed across the RUStance-2023 dataset, where rationale-enhanced prompting substantially outperforms text-only prompting. These findings suggest that jury-generated rationales provide complementary reasoning signals that improve stance classification performance. Deliberative rationales improve performance across all settings, with the largest gains obtained when combined with few-shot learning and fine-tuning Overall, the statistical significance analysis confirms that the improvements obtained from few-shot learning and fine-tuning with jury deliberation rationales are statistically significant and not due to random variations. These findings support the central hypothesis of this work, that structured multi-LLM deliberation provides useful supervisory signals that can improve stance classification when properly integrated into the learning process.

Table 7: Paired bootstrap comparison of model configurations using Macro-F1.

| Comparison | Mean Diff | CI Lower | CI Upper | $p$-value |
|---|---|---|---|---|
| Few-shot vs Zero-shot (Text+Rationale AB) | 0.0367 | 0.0115 | 0.0634 | 0.0026 |
| Fine-tuned vs Few-shot (Text+Rationale AB) | 0.0636 | 0.0178 | 0.1104 | 0.0020 |
| Zero-shot (Text+Rationale AB) vs. Zero-shot (Text) | 0.1557 | 0.0995 | 0.2124 | $< 0.001$ |

## 5.5 Ablation Study: Performance Comparison

**12-Juror vs. 6-Juror Settings.**

As shown in Table 11, A cross-setting comparison revealed that reducing the jury from 12 to 6 members produced a consistent but moderate decline in performance across model classes, highlighting the value of juror diversity and rationale richness for downstream stance modeling. Classical baselines are most affected: the MLP drops from $(F1 = 0.52, Acc = 0.47)$ in the 12-juror setting to $(0.42, 0.42)$ with only six jurors, reflecting the reduced coverage and semantic variability of the rationale embeddings. Zero-shot LLMs show similar sensitivity: Llama-3.3-8B decreases from $(0.45, 0.41)$ to $(0.43, 0.41)$ under text-only prompting, while models conditioned on concatenated Round A or Round A&B rationales exhibit small but systematic reductions of 0.02–0.04 F1. Few-shot prompting mitigates some of this degradation, but still exhibits a measurable gap. For example, the Llama-3.3-8B few-shot performance declines from $(0.60$–$0.65)$ F1 under the 12-juror configuration to $(0.58$–$0.63)$ when only six jurors are available, mirroring a similar contraction for gpt-4.1-mini.

The most notable trend is that while fine-tuned GPT-4.1-mini remains the top-performing model in both settings, its performance also reflects the reduction in available jury signals, decreasing from $(F1 = 0.81, Acc = 0.80)$ with 12 jurors to $(0.69, 0.62)$ in the 6-juror setting. This demonstrates that although fine-tuning is robust to noise and model class variability, it still benefits substantially from the richer deliberation space afforded by a larger LLM jury. Overall, the comparison indicates that (i) more jurors produce more stable and higher-quality rationale embeddings; (ii) few-shot and fine-tuned models leverage these gains most effectively; and (iii) the performance separation between 12- and 6-juror settings widens with model sophistication, underscoring the importance of jury scale for high-fidelity stance annotation and model alignment.

**Self-Consistency Setting**

To compare the proposed multi-LLM jury model with a single-model ensemble strategy, we implemented a self-consistency baseline (Wang et al., 2023), where multiple stochastic outputs are sampled from the same language model and aggregated via majority voting. We used a single LLM (GPT-4.1-mini) and generated independent predictions for each input using a temperature of 0.7 and top-p of 0.95 to encourage output diversity. This process was repeated for $K = 3, 6, 9, 12$, and 24 samples per input, allowing us to compare sampling diversity through self-consistency with model diversity through the multi-LLM jury. For each tweet, the model was instructed to output exactly one stance label, and ties were resolved deterministically by selecting the earliest sampled label among the tied candidates.

The results on the RUStance-2023 dataset are shown in Table 8. Increasing the number of samples improves performance from $K = 3$ to $K = 6$, with accuracy increasing from 0.5667 to 0.5967 and Macro-F1 increasing from 0.5513 to 0.5872. Performance remains nearly unchanged at $K = 9$, where accuracy is still 0.5967 and Macro-F1 slightly increases to 0.5881. However, further increasing the number of samples does not lead to additional gains: at $K = 12$, accuracy decreases to 0.5900 and Macro-F1 drops to 0.5772, with similar performance at $K = 24$. These results suggest that self-consistency provides moderate benefits by aggregating multiple stochastic outputs from the same model, but the gains saturate quickly after $K = 6$ or $K = 9$. Simply increasing the number of samples does not monotonically improve performance. In contrast, the multi-LLM jury setting introduces diversity across heterogeneous models rather than only through stochastic decoding of a single model, which may provide a stronger and more complementary source of variation. This comparison highlights an important distinction between sampling diversity and model diversity. Self-consistency primarily improves performance by reducing the variance in individual model outputs, whereas the proposed jury model leverages systematic disagreement between different models as a signal of ambiguity and uncertainty. The results suggest that model diversity provides a stronger signal than sampling diversity alone, particularly for complex stance detection tasks where subjective interpretation plays a significant role.

Table 8: Performance of self-consistency with different numbers of samples on the RUStance-2023 dataset.

| Method | Accuracy | Macro-F1 | Weighted F1 |
|---|---|---|---|
| Self-Consistency (K=3) | 0.5667 | 0.5513 | 0.5513 |
| Self-Consistency (K=6) | 0.5967 | 0.5872 | 0.5872 |
| Self-Consistency (K=9) | 0.5967 | 0.5881 | 0.5881 |
| Self-Consistency (K=12) | 0.5900 | 0.5772 | 0.5772 |
| Self-Consistency (K=24) | 0.5900 | 0.5764 | 0.5764 |

### 5.6 Failure Analysis

To better understand the limitations of the proposed jury Model, we conducted a qualitative analysis of misclassified examples. We observed that most model errors fell into a small number of recurring categories, such as implicit stance, sarcasm/irony, mixed stance, target confusion, ambiguity boundary, and context dependence, as summarized in Table 9. Detailed examples for each category are provided in Appendix A, Table 10. The most common errors occurred in tweets with implicit stance and sarcasm, where the stance is not explicitly stated and requires contextual interpretation. We also observed frequent errors in multi-target tweets, where the models disagreed on which political actor was the primary stance target. These findings suggest that many classification errors are caused by genuine linguistic ambiguity rather than simple model mistakes, which further supports the central premise of this work that disagreement between models is a signal of uncertainty in subjective NLP tasks.

## 6 Ethical Considerations

This work involves politically sensitive stance labels and LLM-generated rationales. Such labels may reflect biases in the underlying juror models, particularly in contested settings such as the Russia–Ukraine conflict. Because the fine-tuned Judge learns from model-generated rationales, label biases, and systematic juror errors may propagate into downstream predictions. We therefore treat disagreement, uncertainty, and minority rationales as important diagnostic signals rather than as noise.

The released labels and rationales should not be interpreted as verified evidence of an author's beliefs, identity, intent, or political allegiance. LLM rationales are model-generated explanations, not ground-truth justifications. Accordingly, the model is intended for research on annotation methodology, aggregate discourse analysis, and human-in-the-loop decision support. It should not be used as a standalone tool for moderation, surveillance, profiling, or enforcement. Users should preserve uncertainty signals and route high-disagreement or high-stakes cases to human review.

The analogy "12 Angry Men" is procedural rather than psychological or conversational. Round B is not an open-ended debate among agents and does not model human minority-to-majority persuasion. Instead, it is a single re-prompt in which each juror revisits its initial prediction after receiving an anonymized summary of the vote distribution and representative rationales. The title is retained only as a mnemonic for the multi-juror structure, not as a claim of human-like deliberation.

## 7 Conclusion

This study proposes a deliberative *jury-of-LLMs* model for stance detection inspired by *12 Angry Men*. Unlike conventional annotation methods that reduce disagreement to a single consensus label, our approach explicitly preserves divergence through entropy, minority reports, and hung-jury flags. An empirical evaluation of the RUStance-2023 dataset demonstrated that supervised fine-tuning of gpt-4.1-mini achieved the strongest single-model performance ($\approx 0.81$ accuracy), and the jury model provided competitive accuracy with richer interpretability. By comparing human expert annotations, individual LLM baselines, and multi-LLM aggregation, we showed that disagreement can serve as a signal rather than noise, offering diagnostic insights into the uncertainty and subjectivity.

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

## Appendix A: Qualitative Analysis of Model Errors

This appendix presents a qualitative analysis of recurring error patterns observed in the RUStance stance-classification experiments. We focus on representative cases where the expert label is defensible, and the model prediction illustrates a meaningful limitation of the system. Importantly, we do not treat every mismatch between the model prediction and the expert label as an equally clear model failure. Some errors arise from ambiguity in the label schema itself, especially near the boundary between Neutral and Unclear.

In this analysis, *Neutral* refers to a discussion of the conflict in a descriptive, informational, or objective manner without taking sides. The boundaries are used for news reports and factual content that avoid loaded adjectives and narrative framing. *Unclear* refers to Stance cannot be determined due to ambiguity, sarcasm, lack of context, or irrelevance. The boundaries are reserved for posts where even a human expert cannot confidently assign a side due to missing information (Anonymous (2026)).

Table 9: Common failure categories observed in stance classification.

| Failure Type | Description |
|---|---|
| Implicit stance | Stance implied without explicit keywords. |
| Sarcasm/irony | Literal wording opposite to the intended stance. |
| Mixed stance | Multiple political targets in one tweet. |
| Target confusion | Incorrect identification of the stance target. |
| Ambiguity boundary | Neutral statements interpreted as unclear. |
| Context dependence | Requires external knowledge. |

Table 10: Judge prediction errors with interpretations.

| Post (verbatim; tweet ID) | Gold Label | Judge | Revised interpretation |
|---|---|---|---|
| "Sorry #Ukraine, but the More Vlad attacks, the more money I make to pay off 2021 taxes. Unfortunately, this is the perverse system the current administration has established." | Anti-NATO | Unclear | **Sarcasm/irony.** Gold Label is defensible under the target-priority rule because the criticism targets Western policy. Ten of eleven jurors also fail here; a juror's rationale identifies the sarcasm but cannot resolve its target, making this the hardest genuine failure mode. |
| "While #refugees from #Asia and #Africa received a cold response, #Belgium fast-tracked the applications of Ukrainians..." | Anti-NATO | Neutral | **Implicit stance.** Gold Label is defensible because the post criticizes Western double standards without anti-NATO keywords. The Judge reads it as descriptive reporting, missing the implicit anti-Western framing. |
| "Declassified documents show security assurances against #NATO expansion to Soviet leaders from Baker, Bush, Genscher, Kohl..." | Pro-NATO | Anti-NATO | **Context dependence (quotation).** Whether quoting the broken-promise narrative endorses it requires source context. The jury splits four ways, correctly flagging the ambiguity that the Judge resolves in the wrong direction. |
| "Russian Ship, go .... yourself !!! #NATO #USA #UK #GERMANY #POLAND #UKRAINE #PUTIN-HITLER" | Pro-NATO | Pro-Ukraine | **Mixed stance.** The post invokes both the Ukrainian resistance slogan and an alliance frame; Gold Label and model select different primary targets. |
| "I say not another penny to #Ukraine until we can see Zelensky's tax returns. We're paying a lot of money to keep his government in power." | Anti-NATO | Pro-Russia | **Target confusion.** The gold label is reasonable because the post attacks Western or US aid to Ukraine, which fits the Anti-NATO / anti-West policy frame. The model detects anti-Ukraine sentiment but maps it too strongly to Pro-Russia. |

## Appendix B: Prompt Template and Juror Configuration

## A Prompt Template, Label Schema, and Juror Configuration

This appendix documents the prompt template, stance label schema, juror model roster, and persona variants used in the multi-LLM jury experiments.

### Stance Label Schema

All RUStance-2023 experiments use a six-way stance label schema:

- **Pro-Ukraine**: supports Ukraine's actions, sovereignty, or defense.

- **Pro-Russia**: supports Russia's actions, narratives, or justification of the war.

- **Pro-NATO**: supports NATO intervention, presence, expansion, or alliance activity.

- **Anti-NATO**: opposes NATO presence, expansion, intervention, or Western military involvement.

- **Neutral**: discusses the conflict without taking a clear side.

- **Unclear**: stance is ambiguous, sarcastic, irrelevant, or underspecified.

### Juror Model Roster

The jury-of-LLMs configuration consists of 12 jurors: J1 (gpt-4o-mini-2024-07-18), J2 (gemini-2.5-flash), J3 (claude-3.5-sonnet), J4 (grok-4), J5 (llama-3.3-70b-instruct), J6 (llama-3.3-nemotron-super-49b-v1), J7 (deepseek-chat-v3.1), J8 (qwen3-coder), J9 (glm-4.5), J10 (jamba-mini-1.7), J11 (mistral-nemo), and J12 (wizardlm-2-8x22b). This diverse set of models forms the default jury used in the jury-of-LLMs configuration, as summarized in Table 1.

### System Prompt

> You are a careful, literal annotator for stance in short social-media posts. Label EXACTLY one of: [Pro-Ukraine, Pro-Russia, Pro-NATO, Anti-NATO, Neutral, Unclear].

> Definitions: Pro-Ukraine supports Ukraine's actions or sovereignty. Pro-Russia supports Russia's actions or narratives. Pro-NATO supports NATO intervention or alliance. Anti-NATO opposes NATO presence or expansion. Neutral discusses the conflict without taking a side. Unclear means ambiguous or irrelevant in terms of stance.

> Rules: Prefer Unclear when evidence is insufficient, sarcastic, or conflicting. Neutral indicates explicit balance or no clear stance, while Unclear indicates ambiguity, sarcasm, or missing context. Pro-Ukraine and Pro-Russia refer to support, defense, or justification signals; rely only on the text. Pro-NATO and Anti-NATO are about NATO specifically, not generic West or Russia references. Cite short evidence snippets from the text only and do not add external facts. If unsure, set uncertain to true and still choose the best label.

> Output strict JSON with keys: label, confidence, uncertain, rationale, and evidence. No extra commentary.

### Persona Variants

1. Former military strategist, pragmatic and focused on tactical and strategic implications.

2. Human-rights student, focused on humanitarian and ethical aspects.

3. Investigative journalist, skeptical of sources and inclined toward `Unclear` when evidence is weak.

4. Tech entrepreneur, attentive to internet slang, memes, sarcasm, and coded language.

5. Eastern Europe history professor, interpreting posts through regional historical context.

6. Former diplomat, cautious and attentive to nuance, with a tendency toward `Neutral`.

7. Parent, focused on civilian safety and harm to families.

8. Financial analyst, focused on sanctions, supply chains, market effects, and economic warfare.

9. International-law lawyer, focused on sovereignty, war crimes, and treaty obligations.

10. Disinformation data scientist, focused on coordinated messaging, bot-like language, and viral emotional framing.

11. Anti-establishment talk-show host, distrustful of official narratives and attentive to perceived authenticity.

12. Conflict-zone journalist, skeptical of propaganda and inclined toward `Unclear` without concrete evidence.

**Round A Prompt**

Annotate the stance for the following post.

POST:

`"""{text}"""`

Return JSON only.

**Round B Prompt**

Reconsider your label given an anonymous panel summary.

Reference textual evidence only.

PANEL SUMMARY:

`{summary}`

POST:

`"""{text}"""`

Return JSON only.

# Appendix C: Confusion Matrix Analysis of RUStance-2023 Model Performance

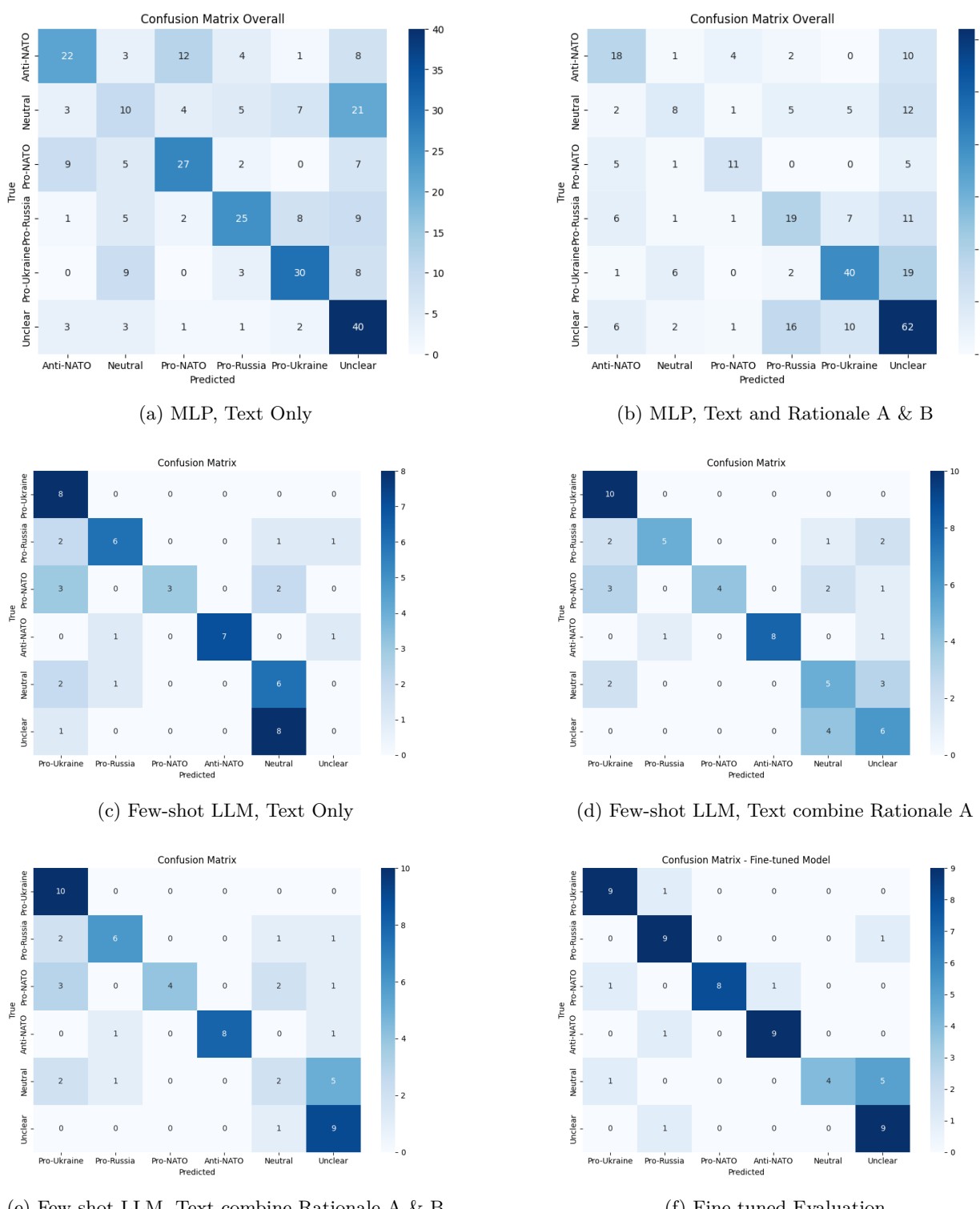

Figure 2: Confusion matrices for the evaluated models.

## Appendix D: Performance comparison for 12-juror and 6-juror

Table 11: Performance comparison for 12-juror and 6-juror settings.

| Jury | Model | F1 | Acc | Input / Prompting |
|------|-------|-----|-----|-------------------|
| 12 | MLP | 0.52 | 0.47 | Text combine rationale A & B |
| 12 | Llama-3.3-8B Zero-shot | 0.45 | 0.41 | Text only |
| 12 | gpt-4.1-mini Zero-shot | 0.51 | 0.51 | Text only |
| 12 | Llama-3.3-8B Zero-shot | 0.54 | 0.51 | Text + rationale A |
| 12 | gpt-4.1-mini Zero-shot | 0.58 | 0.58 | Text + rationale A |
| 12 | Llama-3.3-8B Zero-shot | 0.57 | 0.54 | Text + rationale A &B |
| 12 | gpt-4.1-mini Zero-shot | 0.60 | 0.60 | Text + rationale A &B |
| 12 | Llama-3.3-8B Few-shot | 0.60 | 0.59 | Text combine rationale A & 3 examples |
| 12 | gpt-4.1-mini Few-shot | 0.63 | 0.63 | Text combine rationale A & 3 examples |
| 12 | Llama-3.3-8B Few-shot | 0.65 | 0.65 | Text combine rationale AB & 3 examples |
| 12 | gpt-4.1-mini Few-shot | 0.65 | 0.63 | Text combine rationale AB & 3 examples |
| 12 | gpt-4.1-mini Fine-tuned | 0.81 | 0.80 | Text + rationale A& B |
| 6 | MLP | 0.42 | 0.42 | Text combine rationale A &B |
| 6 | Llama-3.3-8B Zero-shot | 0.43 | 0.41 | Text only |
| 6 | gpt-4.1-mini Zero-shot | 0.51 | 0.50 | Text only |
| 6 | Llama-3.3-8B Zero-shot | 0.56 | 0.54 | Text combine rationale A |
| 6 | gpt-4.1-mini Zero-shot | 0.53 | 0.51 | Text combine rationale A |
| 6 | Llama-3.3-8B Zero-shot | 0.51 | 0.51 | Text combine rationale A & B |
| 6 | gpt-4.1-mini Zero-shot | 0.56 | 0.56 | Text combine rationale A & B |
| 6 | Llama-3.3-8B | 0.63 | 0.62 | Text combine rationale A & 3 examples |
| 6 | gpt-4.1-mini Few-shot | 0.58 | 0.57 | Text combine rationale A & 3 examples |
| 6 | Llama-3.3-8B Few-shot | 0.53 | 0.53 | Text combine rationale AB & 3 examples |
| 6 | gpt-4.1-mini Few-shot | 0.63 | 0.62 | Text combine rationale AB & 3 examples |
| 6 | gpt-4.1-mini Fine-tuned | 0.69 | 0.62 | Text combine rationale A & B |

## Appendix E: Fold-Level Results on RUStance-2023

To assess variability across data splits, we report fold-level performance for the RUStance-2023 experiments evaluated under 4-fold stratified cross-validation. Each fold contains approximately 75 examples. We report Accuracy, Macro-F1, and Weighted-F1 for each fold, together with the mean and sample standard deviation across folds.

| Model | Metric | Fold 1 | Fold 2 | Fold 3 | Fold 4 | Mean ± Std. |
|---|---|---|---|---|---|---|
| Zero-shot LLM (Text) | Accuracy | 0.467 | 0.453 | 0.500 | 0.467 | 0.472 ± 0.020 |
| | Macro-F1 | 0.436 | 0.435 | 0.489 | 0.438 | 0.450 ± 0.027 |
| | Weighted-F1 | 0.433 | 0.435 | 0.493 | 0.437 | 0.449 ± 0.029 |
| Zero-shot LLM (Text + Rationale A) | Accuracy | 0.600 | 0.480 | 0.600 | 0.507 | 0.547 ± 0.063 |
| | Macro-F1 | 0.600 | 0.471 | 0.604 | 0.499 | 0.543 ± 0.069 |
| | Weighted-F1 | 0.602 | 0.472 | 0.605 | 0.498 | 0.544 ± 0.069 |
| Zero-shot LLM (Text + Rationale AB) | Accuracy | 0.667 | 0.573 | 0.627 | 0.560 | 0.607 ± 0.049 |
| | Macro-F1 | 0.666 | 0.570 | 0.642 | 0.560 | 0.609 ± 0.052 |
| | Weighted-F1 | 0.669 | 0.571 | 0.641 | 0.559 | 0.610 ± 0.053 |
| Few-shot LLM (Text + Rationale AB) | Accuracy | 0.707 | 0.640 | 0.667 | 0.600 | 0.653 ± 0.045 |
| | Macro-F1 | 0.693 | 0.633 | 0.674 | 0.590 | 0.647 ± 0.046 |
| | Weighted-F1 | 0.696 | 0.635 | 0.673 | 0.586 | 0.648 ± 0.048 |

Table 12: Fold-level performance on RUStance-2023 under 4-fold stratified cross-validation.

