# OpenReview forum: "‘12-Angry LLMs’ - Divergence from Deliberation as Signal for Complex Stance Detection"
_TMLR — Under review for TMLR_

### Review · Reviewer_LSaD · 2026-06-12

**Summary Of Contributions:**

The paper proposes a "jury of LLMs" framework for stance detection in which twelve heterogeneous LLMs first make independent stance judgments, then enter a second deliberation round when disagreement or low confidence is detected. The resulting votes, rationales, and deliberation traces are aggregated and used as inputs for downstream stance classifiers, including MLP baselines, zero/few-shot LLM prompting, and a fine-tuned GPT-4.1-mini Judge. The paper evaluates the approach primarily on a 300-instance RUStance-2023 dataset and reports improvements from adding Round A/B rationales, with the strongest reported RUStance performance around 0.81 accuracy / 0.79 macro-F1. It also reports experiments on PStance, SemEval-2016, and GWStance, and proposes releasing a dataset with expert labels and jury deliberation traces.

**Audience:**

Yes

**Audience Explanation:**

I think the annotation pipeline may lie in the interests of some audience.

**Claims And Evidence:**

Yes

**Claims Explanation:**

I think the empirical results can justify the claims made in the manuscript.

**Requested Changes:**

Only one minor concern on the manuscript. The appendix failure analysis is useful, but some cases are not fully convincing. First, several errors appear to hinge on an underdefined boundary between Neutral and Unclear; without clearer annotation guidelines, it is hard to know whether these are model mistakes or reasonable responses to an ambiguous label schema. Second, some examples seem relatively easy for modern strong LLMs under a well-specified prompt, so the authors should provide exact prompts/outputs and more representative hard cases. Otherwise, the failure analysis may overstate the difficulty of these examples or conflate model limitations with prompt/schema ambiguity.

---

> ### Author Response · Authors · 2026-07-10
>
> We revised the appendix A failure analysis to avoid presenting all model gold mismatches as equally clear model failures. The revised appendix now explicitly defines the "Neutral/Unclear" distinction and explains that some such cases should be interpreted as schema-boundary ambiguity rather than as unambiguous model error. We also replaced weak examples with RUStance finetune error results examples and now report the post-text, gold label, judge prediction, and revised interpretation. The revised examples focus on representative hard cases: sarcasm/irony, implicit stance, mixed-target stance, target confusion, and context dependence. (Added the prompt in Appendix B)

---

### Review · Reviewer_hjZ2 · 2026-06-25

**Summary Of Contributions:**

This study proposes a new method for stance detection. The method consists of an expert LLM judge and a multi-turn deliberation process that aggregates votes from a group of independent LLM "jurors" (rather than using a single judge or simple aggregations over multiple judges, such as majority voting). The rationale is that this makes effective use of the diversity of the LLMs and should improve the reliability of the final judgement.

The jury-based approach is general enough to be applied to any domain, though this is not highlighted prominently in the paper.

The approach is evaluated against other stance detectors (MLPs, zero-shot and fine-tuned LLMs) and human judgements across three stance detection datasets, with good in-domain but weak out-of-domain results.

The study also promises the release of a dataset of social media posts on the Russia–Ukraine conflict annotated with the stance labels and deliberation processes produced by the proposed method.

**Audience:**

Yes

**Audience Explanation:**

Stance detection is an instance of a wide array of subjective tasks where judges or annotators inherently disagree and reliable judgements are difficult to obtain. Beyond the stance detection community itself, there is likely a broader audience interested in these findings. Potentially any researcher using an LLM judge could be interested in this paper to create a more reliable automatic annotation system.

(That said, testing the approach on other tasks would be required to substantiate the claim that its applicability extends beyond stance detection.)

**Broader Impact Concerns:**

It would be desirable to discuss the implications of deploying these methods at scale on real-world data, and in particular the consequences of errors. In this context, the value of having access to deliberation traces would also be worth addressing. Additionally, it would be worth discussing the applicability of this work to other subjective judgement domains.

The paper does not contain a Broader Impact Statement.

**Claims And Evidence:**

No

**Claims Explanation:**

1. The second deliberation round (round B) only activates under certain uncertainty conditions, which are underdefined. In particular, how "reliability" is measured is not explained in detail. The Dawid & Skene approach to aggregation, which I believe is the source of the reliability scores, is described only very briefly. This is problematic, as the uncertainty determination is central to the method proposed.
2. How the judge is trained is also insufficiently explained, and it is diluted across sections. Section 3.3 merely states that the model is trained "to emulate expert consensus" and to "synthesise deliberation traces", without specifying where the gold labels come from. Section 4.1 clarifies that the training data consists of LLM juror deliberations with gold human-annotated labels attached. *If this is the case, it creates a disconnect between the deliberation process and the outcome*: there could be instances where most LLM jurors vote for a certain label but the expert annotation disagrees. The paper claims this process "ensures that the fine-tuned Judge model learns how human experts resolve disagreements among jurors", however, the input deliberations are LLM-generated, not ones that human judges have actually adjudicated.
3. Interpretation of results, section 5.1: "achieving the highest Macro-F1 on three out of four datasets" -- the proposed method is _not_ the top performer on two out of four datasets, and for PStance the margin over the second best is only 0.003. The clear advantage is only on RUStance, the dataset on which the judge is trained, which drives the average Macro-F1 across datasets. This indicates weak generalisation.
4. Interpretation of results, section 5.1, continued: "Few-shot learning with rationales consistently outperformed zero-shot prompting, and both outperformed classical MLP baselines" -- this is inaccurate: for PStance, MLP outperforms zero-shot, and for GWStance, MLP is the second best approach.
5. Interpretation of results, section 5.3: Again: "Few-shot and fine-tuned LLMs generally outperform MLP baselines." Now this is false for two out of three datasets in this section. Please correct these claims.
6. It is not clear whether statistical significance testing is conducted only on RUStance-2023 or on all datasets.
7. It is not clear whether the fine-tuned judge in the 6-juror setting is the same model fine-tuned on 12 jurors. If so, the interpretation that "the performance separation between 12- and 6-juror settings widens with model sophistication" may not be warranted.
8. No statistical significance results are reported for the 12 vs 6 juror ablation.

**Requested Changes:**

1. *Address the points raised under "Are the claims made in the submission supported by accurate, convincing and clear evidence?"*
2. In the second-to-last paragraph of the introduction, the work is framed as advancing "the RUStance-2023 study" through its main contributions. This is too narrow: the contributions do extend to other datasets and domains. Please rewrite this paragraph accordingly.
3. Please cite relevant LLM-as-a-judge work; currently no such references appear [Zheng et al., 2023; Liu et al., 2023; Chiang and Lee, 2023; Fu et al., 2024; Bavaresco et al., 2025; Wang et al., 2025; etc.]
4. The choice of 12 judges is motivated by a film title and the size of a UK courtroom jury. Neither provides any guarantee that 12 is the right jury size for LLMs and the paper does include a small ablation for this variable. However, the composition of the jury is also likely to matter. Please study more variations in both size and composition. Moreover, training the judge on a variable number of jurors would likely further improve generalisation and practical usability.
5. Please discuss the computational cost of the proposed approach relative to the baselines and other standard alternative approaches. The method involves not only multiple judges but also a fine-tuned meta-judge. Even setting fine-tuning aside, please report the token count or wall-clock time required compared to majority voting and a single judge with self-consistency.
6. Please increase the number of samples in the self-consistency analysis. In the multi-agent setting there are 12 (or 6) distinct models; in the self-consistency analysis there are only 12 (or 3?) generations from the same model. This is not a fair comparison.
7. Please fix the bolding in Table 3: bold the best result per column rather than always highlighting the proposed method. The proposed method is not the top performer on two out of four datasets, and on one it leads the second-best by only 0.003.
8. Please remove Table 5, which is entirely subsumed by Table 3.
9. p. 4, Section 3.2, first paragraph: "the six-class schema" — this schema has not yet been introduced at this point; please fix.
10. Please define reliability, entropy, and margin before they are used to describe round B.
11. Section 2: the heading should read "Related Work", not "Related Works"
12. p. 2: 'judge' is missing a closing quotation mark.
13. p. 4, Section 3.2, first paragraph: "the six-class schema" — the schema has not yet been introduced; please fix.

---

> ### Author Response · Authors · 2026-07-10
>
> **Answer to the problem**
>
> **P1.** We have expanded the method section 3.2  to clarify when Round B is activated and how reliability is computed. Round B is not always executed; it is triggered when the Round A vote distribution indicates uncertainty, such as high entropy, a small vote margin between the top labels, or low agreement with the provisional final label. Reliability is defined as the fraction of jurors whose votes agree with the final aggregated label and is used as an internal, agreement-based confidence measure rather than a measure of correctness. We also clarified the aggregation methods. Majority voting selects the most frequent juror label. Dawid-Skene treats jurors as noisy annotators and estimates their label-specific reliability patterns, using these estimates to infer the most likely latent label. We now report majority voting and Dawid-Skene as separate jury-only baselines, distinct from the fine-tuned Judge.
>
> **P2.** We have revised Section 3.3 and added a dedicated subsection that explains the judge training and inference pipeline in greater detail. The fine-tuned judge is not trained on human-authored adjudications of juror disagreements, and we do not claim that human annotators directly reviewed or resolved each LLM deliberation. Rather, the judge is trained on LLM-generated deliberation traces paired with independently obtained gold-standard human stance labels.
> Accordingly, we have revised Section 4.1 to avoid suggesting that the model learns an explicit human disagreement-resolution process. Our intended claim is more limited: the judge learns a supervised mapping from the tweet text and LLM-generated rationales to the gold human annotation. This supervision can help the model identify cases where the jury’s reasoning aligns with the expert label, as well as cases where the majority vote or rationale pattern may be misleading.
>
> **P3.** We agree that the previous wording overstated the proposed method's cross-dataset generalization. In the revised manuscript, we have removed the claim that the fine-tuned Judge achieves the highest macro-F1 on three out of four datasets. We have revised Section 5.1 to correct the claim. The proposed fine-tuned judge does not achieve the highest macro-F1 on three out of four datasets. In the corrected results, it is best on two datasets: RUStance-2023 and PStance. The Few-shot LLM performs best on SemEval, and the MLP with rationale augmentation performs best on GWStance. We now state this explicitly and avoid claiming uniform superiority. We also agree that the PStance margin is small. The fine-tuned Judge obtains 0.946 Macro-F1 on PStance, while the few-shot LLM obtains 0.943, a difference of only 0.003. We therefore no longer interpret P-stance as strong evidence of a clear advantage. Instead, we describe the judge as having the highest average Macro-F1, with its clearest advantage on RUStance-2023. To avoid overstating generalization, we have revised the interpretation of Table 3.
>
> **P4.** In the revised Section 5.1, we have reorganized the claims. The revised interpretation now states that performance is dataset-dependent. In particular, while rationale-augmented LLM methods perform strongly on RUStance-2023 and PStance, the MLP baselines remain competitive in other settings: on PStance, MLP with rationale augmentation outperforms the zero-shot text-only setting, and on GWStance, MLP with rationale augmentation achieves the best macro-F1 score overall. We now report that the fine-tuned Judge achieves the strongest average macro-F1 across datasets, but it is not uniformly the best method on every dataset.
>
> **P5.**  We thank the reviewer for pointing this out. In the revised manuscript, we have removed the former Section~5.3 subsection to avoid duplication and to prevent overgeneralizing from dataset-specific results. As a result, the claim that few-shot and fine-tuned LLMs generally outperform MLP baselines no longer appears there. The relevant results are now reported and discussed in Section 5.1, where performance differences are described in a dataset-specific manner rather than as a broad claim across all datasets.
>
> **P6.** Statistical significance testing was conducted only on the RUStance-2023 dataset.
>
> **P7.** The fine-tuned Judge in the 6-juror setting is not the same model fine-tuned on 12-juror inputs. We trained and evaluated the 6-juror judge using 6-juror inputs and the 12-juror judge using 12-juror inputs. Thus, the comparison reflects each setting under its corresponding input format rather than applying a 12-juror-trained judge to 6-juror data.

---

> ### Author Response · Authors · 2026-07-10
>
> **P8.** Because this analysis is intended as an architectural/input-size ablation rather than the primary hypothesis test. Our statistical testing is restricted to the main RUStance-2023 comparisons, while the 6-juror setting is included to provide descriptive evidence about how performance changes when the number of jurors is reduced. Therefore, we report the performance differences but do not make a formal claim of statistical significance for the 12-vs-6 ablation.
>
> **Requested Changes:**
>
> **RC1.** We have addressed all of the points raised under "Are the claims made in the submission supported by accurate, convincing and clear evidence?”
>
> **RC2.** We have revised the second-to-last paragraph of the Introduction to clarify the contribution. We also added one paragraph after the summary of contributions to indicate a broader contribution of the paper. It states: “Overall, our work introduces a general deliberative annotation and evaluation model for stance detection. By combining human expert gold labels with LLM-based jury deliberation, we provide a broader benchmark for studying disagreement as a cue in stance detection."
>
> **RC3.** We thank the reviewer for this suggestion. Although we disagree that our work falls under the umbrella of LLM-as-a-judge, where LLMs are used for evaluating AI-generated texts rather than (imprecise) text classification problems, in the revised manuscript, we have added citations to a few LLM-as-a-judge and LLM-based evaluation works. We have added these references to the Introduction section.
>
> **RC4.** We agree that twelve jurors should not be interpreted as the optimal LLM jury size. In the revised manuscript, we clarify that twelve is a fixed experimental design choice and a mnemonic for the multi-juror setting, not a theoretically justified optimum.
> We have a 6-vs-12-juror ablation and new RUStance-2023 controls in section 5.3 to better examine the effects of jury size and composition. The 6-juror ablation tests whether reducing the number of jurors affects performance. The single-model twelve-rationale control keeps the number of rationales fixed but removes cross-model diversity. In that control, the fine-tuned judge achieves only 0.597 accuracy, 0.566 macro-F1, and 0.567 weighted-F1, suggesting that simply adding more rationales from a single model does not reproduce the benefits of the heterogeneous jury. We also add a length-matched consensus-rationale control, which achieves 0.740 accuracy, 0.725 macro-F1, and 0.725 weighted-F1, showing that rationale quantity helps but does not fully explain the jury-based gains. (Juror configuration in Table 1 and Appendix B was added for prompt design.)
>
> **RC5.** Because wall-clock time was not consistently logged during the original experiments, we do not report runtime estimates. Instead, we report the available token counts and discuss relative computational scaling. The RUStance-2023 dataset used in our experiments contains 300 Twitter/X posts. In the 4-fold Judge fine-tuning setup, each fold used approximately 269,752 training tokens, for a total of 1,079,008 training tokens across four folds. Using an illustrative fine-tuning price of USD 5 per million training tokens, the Judge fine-tuning stage would cost approximately USD 5.40.
> The jury-generation stage has a different cost profile because it uses multiple LLMs, and token prices vary by model provider, model size, and input or output token type. For RUStance-2023, the 12-juror setup consumed approximately 8.91 million tokens across Round A and Round B deliberations. We therefore report this as token usage rather than a single fixed dollar cost. Under a simplified illustrative rate of USD 5 per million tokens, this would correspond to approximately USD 44.55, but the actual cost may be higher or lower depending on the specific juror models and pricing schedule used. The main computational cost, therefore, lies in generating multi-juror deliberation traces rather than in fine-tuning the judge, and this cost scales approximately linearly with the number of posts, jurors, and deliberation rounds.
>
> **RC6.** In the revised manuscript, we expanded the self-consistency analysis on RUStance-2023 by evaluating K = 3, 6, 9, 12, and 24 samples from the same backbone model.
>
> **RC7.** We have corrected the bolding in Table 3 so that the best value in each column is bolded.
>
> **RC8.** We have removed Table 5 from the revised manuscript. To avoid duplication, we now report the relevant overall performance results only in Table 3 and refer to that table in the accompanying discussion.
>
> **RC9.** We have revised the sentence to avoid referring to the ``six-class schema'' before it is introduced. The text now refers more generally to the predefined stance label set.

---

> > ### Author Response · Authors · 2026-07-10
> >
> > **RC10.** We have revised Section 3.2 to define entropy, margin, and reliability before they are used to describe the Round B trigger. Entropy is computed over the Round A vote distribution; margin is the difference between the top two label proportions; and reliability is the proportion of jurors supporting the final aggregated label. We also clarify that reliability is an internal, agreement-based confidence measure, not a correctness measure against the gold label.
> >
> > **RC11.** We have corrected the Section 2 heading from “Related Works” to Related Work' in the revised manuscript. Thank you for the correction.
> >
> > **RC12.** We have added the missing closing quotation mark after ``judge''
> >
> > **RC13.**  We have revised the sentence to avoid referring to the ``six-class schema'' before it is introduced. The text now refers more generally to the predefined stance label set.
> >
> > **Broader Impact Statement.**
> >
> > We have added a new Section 6 on ethical considerations, which discusses label bias propagation, uncertainty, and appropriate-use limitations. The statement clarifies that the system should not be used as a standalone tool for moderation, surveillance, or enforcement and that LLM rationales should be treated as model-generated explanations rather than verified evidence.

---

> > > ### Author Response · Authors · 2026-07-13
> > >
> > > Dear Reviewer,
> > >
> > > Thank you for providing valuable suggestions for the improvement of our paper. We have done extensive revision of the paper and hope all of your feedbacks are answered satisfactorily.
> > >
> > > We kindly requesting you for review the revision and update your review accordingly. We would love to response if you have any further queries and suggestions.
> > >
> > > Thank you again.

---

### Review · Reviewer_FNPD · 2026-06-28

**Summary Of Contributions:**

The paper proposes a pipeline for stance detection that treats annotator disagreement as signal, rather than noise. 12 LLMs are considered as independent "jurors" and put through 2 rounds: round A is to independently cast a stance label. If the aggregated vote shows low reliability then round B is triggered for each juror to revisit the post, given an anomylized summary. Votes are aggregated, then the rationale with gold label are used to fine-tune a final Judge model. Experiments cover RUStance2023 and balance 300-instance subsets of PStance, SemEval2016 task 6 and GWStance.

**Additional Comments:**

The "12 Angry Men" framing is engaging but oversells the mechanism: Round B is a single re-prompt with an anonymized vote summary (self-refine-with-peer-context), not the reasoned minority-to-majority persuasion the analogy implies.

**Audience:**

Yes

**Audience Explanation:**

The disagreement as signal framing and the LLM-jury construction could be interesting, given the findings can be proven more clearly.

**Broader Impact Concerns:**

I do not see grounds to withhold the paper on ethics, but a short Broader Impact Statement addressing label-bias propagation and appropriate-use limitations should be added.

**Claims And Evidence:**

No

**Claims Explanation:**

The core idea of juries is interesting, however it has appeared for a long time [1]. This makes me question the paper's novelty, where the authors stated they "introduce a new methodology". I notice the authors referred to [1], but I am confused why the authors did not include that baseline in, as it is one of the main competitors. Additionally, I find many central claims are not supported, and some directly contradicted by the paper's own tables. I will group the problems by severity below.

1. **Headline results contradict**: Authors mentioned Judge achieves the highest macro F1 on 3 out of 4 datasets, but in table 3 the authors bold all results of their method, but for SemEval and GWStance the results are not best. Furthermore, the Avg F1-macro (last column) of the authors' method is wrongly calculated. I highly suspect there is something wrong with the way the authors report the results. Please explain clearly.

2. **Cross-domain transfer problems**: Table 5 reports numbers on OOD data, and the results show authors' method achieves best, which is good; however, the main point I care about is how can the schema help is not clearly analyzed. Furthermore, as far as i understand, the results of the authors' method was re-run and the Judge is fine-tuned per dataset? If so, it directly contradicts the statement in Abstract "F1 0.94 on the out-of-domain PStance dataset using few-shot prompting"?

3. Jury's own prediction is never added. It should be a straight forward baseline. Without this it is hard to evaluate whether the boost in performance come from the author's method design, or simply from adding more reasoning text

4. The claim of the paper mentioned disagreement of the jurors is a signal. However, the experiments only show adding rationale text helps, which aligns with CoT enrichment findings. I see no control that holds rationale text fixed while varying disagreement: e.g. (i) twelve rationales from a single model (no model disagreement) into the same Judge, and (ii) one length-matched consensus rationale to rule out "more CoT helps." Table 8's self-consistency gestures at this but uses GPT-4o-mini, not the Judge's GPT-4.1-mini backbone, so it isn't controlled. As stands, divergence-as-signal is asserted, not demonstrated.

5. The paper contains a lot of hyper parameters (margin, entropy and reliablity). I see no ablation on that, and no explanation why the authors fixed those numbers.

**References**
[1] Replacing Judges with Juries: Evaluating LLM Generations with a Panel of Diverse Models - arxiv

**Requested Changes:**

I tag each item as [Critical] or [Would strengthen]

[Critical] Add the baseline [1] and reposition the novelty claim. Since the jury-of-LLMs idea is already established in [1] — which the authors cite — please include it as a baseline and soften the "we introduce a new methodology" framing, making explicit what this work adds beyond [1].

[Critical] Fix and explain the Table 3 results. Please correct the bolding so only the best result per dataset is bolded (the authors' method is not best on SemEval and GWStance), recompute the Avg Macro-F1 column for the proposed method (the current value does not match the row), and reconcile the "highest on 3 out of 4 datasets" statement with the actual table (it is 2 out of 4). A clear explanation of how the numbers were produced is needed, as the abstract's headline F1 depends on them.

[Critical] Clarify the cross-domain setup and fix the abstract accordingly. Please state explicitly whether the Judge is fine-tuned per dataset. If so, the abstract claim "F1 0.94 on the out-of-domain PStance dataset using few-shot prompting" is contradictory and must be reworded, since per-dataset fine-tuning is not out-of-domain few-shot transfer. In addition, please add an analysis of how the six-class schema transfers to the OOD datasets — at present the strong Table 5 numbers are reported without explaining the mechanism.

[Critical] Add the jury's own prediction (MV / Dawid–Skene) as a baseline. This is a straightforward and essential comparison; without it, it cannot be determined whether the gains come from the method's design or simply from adding more reasoning text. Please also state which aggregation underlies each reported result.

[Critical] Add a control that isolates disagreement from rationale quantity. To support the central claim that disagreement is the signal, please hold rationale text roughly fixed while varying disagreement: (i) twelve rationales from a single model (no cross-model disagreement) into the same Judge, and (ii) one length-matched consensus rationale to rule out "more CoT helps." Please also rerun the self-consistency comparison (Table 8) on the Judge's own backbone (GPT-4.1-mini) rather than GPT-4o-mini, so it is a controlled diversity comparison.

[Would strengthen] Report per-fold mean and std for the main results, given the small (300-instance, ~40–50 test) splits on which the strongest numbers rely.

---

> ### Author Response · Authors · 2026-07-10
>
> **Requested Changes (RC):**
>
> **RC1.** We have now (a) softened the novelty framing throughout the revised manuscript, (b) added a panel-of-LLM-evaluators baseline in the spirit of [1], and (c) made explicit in the text and now also empirically what this work adds beyond [1]. [1] uses a panel of diverse LLMs as independent evaluators and pools their votes to evaluate generated model outputs. Our Jury MV (Round A) baseline follows the same core idea in the stance-classification setting: each juror votes independently once, and the majority label is returned. Our full method goes beyond PoLL by adding disagreement-triggered Round B revisiting, preserving minority rationales and uncertainty traces, and using the Round A/B rationales to fine-tune a downstream judge.
> Hence, the closest operational analog to [1] in our stance-detection setting is the newly added Jury MV (Round A) baseline in Table 3. Here, a diverse panel votes independently in a single round, and predictions are aggregated by majority vote. We also report Jury DS (Round A) as a reliability-aware aggregation variant. Jury DS/MV (Round A) is the final stance label produced by applying Dawid–Skene (DS) / Majority Vote (MV) aggregation to the independent Round-A votes of the 12 LLM jurors.
>
> **RC2.** **a) Average Macro-F1.** The reported Avg Macro-F1 of 0.791 for the fine-tuned judge was a transcription error. The value 0.791 was the RUStance-only Macro-F1 from Table 4 and was mistakenly pasted into the average column of Table 3. The correct unweighted average for the Judge row is computed from the four dataset-level Macro-F1 scores: 0.815, 0.946, 0.693, and 0.911. This gives an average macro-F1 of 0.841. We verified the average macro-F1 values for all other rows in Table 3: 0.654, 0.730, 0.666, 0.757, and 0.777. These values were correct, so the error was confined to a single cell in the Judge row.
>
> **b) Bolding and “3 out of 4” claim.** We have corrected the bolding so that only the best result in each dataset column is bolded. The fine-tuned Judge is best on RUStance2023, with Macro-F1 0.815, and PStance, with Macro-F1 0.946. The few-shot LLM is best on SemEval, with Macro-F1 0.736, and the MLP with rationales is best on GWStance, with Macro-F1 0.929 (the proposed method is 2nd highest). We have therefore revised the text. The Abstract and §5.1 have been updated accordingly.
>
> **RC3.** **a) The setup, stated explicitly (revised section 4.1)**. The reviewer's understanding is correct: the pipeline was applied end-to-end for each dataset. For each of PStance, SemEval, and GWStance, the jury generated Round A/B rationales on that dataset's texts under that dataset's native label schema (favor/against; FAVOR/AGAINST/NONE; agree/disagree/neutral), and every downstream configuration, including the fine-tuned judge, was trained and evaluated on the same dataset under 4-fold stratified cross-validation, with few-shot exemplars drawn from each fold's training split. We have deleted Section 5.3, including Table 5, because it duplicated the PStance, SemEval, and GWStance results already summarized in Table 3, Section 5.1.
>
> **b) A corrected abstract.** The revised abstract sentence now reads: "The pipeline generalizes across stance domains: applied end-to-end to PStance, SemEval-2016, and GWStance; jury-generated rationales improve several downstream configurations, most notably on PStance, where few-shot prompting and the fine-tuned Judge both reach Macro-F1 ≈ 0.94. "
>
> **c) How the schema and rationales help mechanism analysis.**
> Rationales convert implicit cues into explicit evidence. By requiring twelve jurors to justify their decisions, the system surfaces sarcasm, target references, and framing that the downstream model would otherwise need to infer on its own. As a result, the judge operates not only on the original text but also on an organized inventory of evidence, which explains why rationale-augmented rows outperform text-only rows in the majority of the settings. Diversity further strengthens the procedure by turning disagreement into an uncertainty signal. Because the jurors are heterogeneous, their agreement becomes informative, and their disagreement helps identify difficult cases. This signal is used twice: first, to decide when Round B should be triggered, and second, to indicate to the judge why an item is challenging. We clarify that we do not transfer the RUStance six-class ontology unchanged to other datasets; instead, we transfer the jury protocol while using each dataset’s native label verbalizers. We have added new experiments with the RUStance-2023 dataset to understand the schema of rationales and added a new section 5.3 with new results.

---

> > ### Author Response · Authors · 2026-07-10
> >
> > **RC4.** We agree with the reviewer that the jury's own prediction is an essential baseline. We have added jury aggregation baselines to the main results tables, 3 and 4. Specifically, we now report Majority Vote (MV) and Dawid–Skene (DS) aggregation for both the independent Round-A votes and the post-deliberation Round-B votes. The new results show that direct jury aggregation is a meaningful but weaker baseline than the final judge. Jury MV (Round A) obtains an average Macro-F1 of 0.615, Jury DS (Round A) obtains 0.629, Jury MV (Round B) obtains 0.628, and Jury DS (Round B) obtains 0.624. In contrast, the fine-tuned LLM judge using text and Round-A/B rationales achieves an average Macro-F1 of 0.841. This indicates that the final gains are not simply due to taking the jurors’ majority label; rather, the judge benefits from the rationale-conditioned supervised learning step. Most reported jury-based results use majority-vote (MV) aggregation. Dawid–Skene aggregation is used only for rows explicitly labeled “DS.” We have also added Section 5.3 as per your suggestion of RC.5 to check the contributions of divergence against the quantity of rationale texts. This shows that adding more reasoning text does not improve as much as it gains due to divergence signals.
> >
> > **RC5.** We agree with the reviewer that the original submission did not fully separate the effect of disagreement from the effect of rationale quantity. We have therefore added the requested controls. These control experiments are reported on the RUStance setting in Section 5.3 (Disagreement vs. Rationale Quantity Controls) of the revised manuscript.
> > We rerun the self-consistency comparison using the Judge backbone, GPT-4.1-mini, and increase the number of samples up to K=24. The results remain flat: Macro-F1 is 0.5897 for K=3, 0.5872 for K=6, 0.5881 for K=9, 0.5896 for K=12, and 0.5764 for K=24. Thus, simply drawing more samples from the same model does not close the gap to the full heterogeneous jury setting. (Revised Table 8 and Section 5.5 Self-Consistency Setting)
> >
> > Together, these controls show that rational quantity does contribute useful signals, especially in the length-matched consensus condition. However, rational quantity alone does not explain the full performance of the proposed method. The full multi-LLM jury judge achieved 0.815 Macro-F1 on the same RUStance setting, outperforming both the single-model twelve-rationale control and the length-matched consensus-rationale control. We therefore revise the central claim to be more precise: the gains are not due to disagreement alone but to disagreement-aware, multi-perspective rationale supervision that provides additional signal beyond merely adding more reasoning text.
> >
> > **RC6.** In the revised manuscript, we have added an appendix E, which reports the per-fold mean and standard deviation for the main results.
> >
> > **Broader Impact Concerns.**
> >  We have added a new Section 6 on ethical considerations, which discusses label bias propagation, uncertainty, and appropriate-use limitations. The statement clarifies that the system should not be used as a standalone moderation, surveillance, or enforcement tool, and that LLM rationales should be treated as model-generated explanations rather than verified evidence.
> >
> > **Additional Comments.**
> > We agree that the “12 Angry Men” analogy can overstate the mechanism if interpreted literally. The analogy is procedural rather than psychological or conversational. Round B is not an open-ended debate among agents and does not model human minority-to-majority persuasion. It is a single re-prompt in which each juror revisits its initial prediction after receiving an anonymized summary of the vote distribution and representative rationales. The title is retained only as a mnemonic for the multi-juror structure, not as a claim of human-like deliberation. We have added this in the revised manuscript as an ethical statement in section 6.

---

> > > ### Author Response · Authors · 2026-07-13
> > >
> > > Dear Reviewer,
> > >
> > > Thank you for providing valuable suggestions for the improvement of our paper. We have done extensive revision of the paper and hope all of your feedbacks are answered satisfactorily.
> > >
> > > We kindly requesting you for review the revision and update your review accordingly. We would love to response if you have any further queries and suggestions.
> > >
> > > Thank you again.